# Vertigo in Acute Stroke Is a Predictor of Brain Location but Is Not Related to Early Outcome: The Experience of Sagrat Cor Hospital of Barcelona Stroke Registry

**DOI:** 10.3390/biomedicines10112830

**Published:** 2022-11-06

**Authors:** Angela d’Annunzio, Adrià Arboix, Luís García-Eroles, María-José Sánchez-López

**Affiliations:** 1Department of Neurology, Hospital Universitari Sagrat Cor, Quirónsalud, Universitat de Barcelona, 08029 Barcelona, Catalunya, Spain; 2Medical Library, Hospital Universitari Sagrat Cor, Quirónsalud, Universitat de Barcelona, 08029 Barcelona, Catalunya, Spain

**Keywords:** brainstem, cerebellum, cerebral infarction, cerebrovascular diseases, dizziness, imbalance, stroke, stroke registry, vertebrobasilar insufficiency, vertigo

## Abstract

Background: Vertigo is an uncommon symptom among acute stroke victims. Knowledge about the clinical profile, the brain location, and the early outcome in stroke patients with cerebrovascular diseases and vertigo remains limited. Objectives: In this study, the effects of vertigo on cerebral topography and early prognosis in cerebrovascular diseases were investigated. Methods: A comparative analysis in terms of demographics, risk factors, clinical characteristics, stroke subtypes, cerebral and vascular topography, and early outcome was performed between patients with presence or absence of vertigo on a sample of 3743 consecutive acute stroke patients available from a 24-year ongoing single-center hospital-based stroke registry. Results: Vertigo was present in 147 patients (3.9%). Multiple logistic regression analysis showed that variables independently associated with vertigo were: location in the cerebellum (OR 5.59, CI 95% 3.24–9.64), nausea or vomiting (OR 4.48, CI 95% 2.95–6.82), medulla (OR 2.87, CI 95% 1.31–6.30), pons (OR 2.39, CI 95% 1.26–4.51), basilar artery (OR 2.36, CI 95% 1.33–4.17), ataxia (OR 2.33, CI 95% 1.41–3.85), and headache (OR 2.31, CI 95% 1.53–3.49). Conclusion: The study confirmed that the presence of vertigo was not related with increased in-hospital mortality or poor prognosis at hospital discharge. Vertigo is mainly related to non-lacunar vertebrobasilar stroke with topographic localization in the cerebellum and/or brainstem.

## 1. Introduction 

According to WHO data, stroke is one of the predominant causes of mortality and disability worldwide, being the second leading cause of death in adult males and the first in adult females worldwide. Furthermore, stroke is one of the most important causes of disability and dementia in adults [1], leaving up to 50% of patients with chronic disability, which has a huge impact on health care and the economy. These impact data are expected to increase due to the aging of the population.

Age- and sex-standardized rates of stroke mortality have declined in recent decades; however, the absolute number of stroke sufferers, stroke survivors, and overall stroke conditions are large and increasing [2,3,4,5,6].

Hemiparesis and other types of motor weakness, sensory disturbances, visual symptoms, and aphasia or speech disturbances are the most common manifestations of cerebrovascular diseases. In contrast, vestibular syndromes and vertigo are uncommon clinical features in acute stroke patients [7,8,9].

Stroke has a complex pathophysiology. The most recent studies suggest that the brain is exquisitely sensitive to even short-duration ischemia, implicating multiple mechanisms in the tissue damage that results from cerebral ischemia [10,11,12,13,14]. Ischemic stroke initiates a cascade of events that generate ATP depletion, ionic dysregulation, increased glutamate release, and excessive free radical production, as well as edema and inflammation, all of which ultimately contribute to cell death [15,16,17]. In contrast, in intracerebral hemorrhage, the primary cause of injury is oppression and destruction of brain tissue by the hematoma, although the inflammation, coagulation response, and toxicity of released hemoglobin also play a key role [18,19,20]. 

Vertigo is an unpleasant distortion of static gravitational orientation and is defined as a specific type of dizziness consisting of a perception of spinning or tilting with nausea/vomiting and gait unsteadiness. It is the third most commonly presenting major symptom in general medicine clinics and accounts for 3%–5% of all visits [21]. 

Vertigo is usually due to acute peripheral vestibulopathy. However, the prevalence of central vertigo, especially due to acute stroke, is not negligible, accounting for 3–4% of vertigo cases in a clinical study [22].

It should be noted that dizziness/vertigo has also been recognized as a manifestation of epileptic seizures, migraine, or demyelinating disease [23]. Unpleasant vegetative effects, such as nausea and vomiting, associated with vertigo are related to clinical activation of the medullary vomiting center. Making this discrimination is considered a clinical challenge since, for example, acute vertebrobasilar strokes may present with analogous symptoms which mimic an acute vestibular syndrome [24].

The course and prognosis of vertigo syndrome in acute stroke patients are variable. The diagnosis of vertigo in patients presenting with posterior circulation acute stroke has recently increased markedly [25,26]. 

However, little is known about the cerebral location, clinical profile, prevalence in the different stroke subtypes, and early outcome in acute cerebrovascular patients with vertigo. 

Most central ischemic vertigo syndromes are secondary to paramedian or lateral tegmental infratentorial lesions [27]. However, the specific sites of brain lesions and arterial vessel disruption that cause the characteristic vestibular syndromes or vertigo are less well known [24,25,26,27]. 

Furthermore, the clinical spectrum of acute stroke and vertigo includes brain-stem and cerebellar signs and symptoms, but vertigo has also long been recognized as an isolated manifestation of anterior inferior cerebellar artery ischemic stroke or in small insular acute strokes [27]. 

Previous studies have analyzed the presence of vertigo in a sample of acute strokes in general, as well as in ischemic or hemorrhagic strokes [22]. However, the study and data on the frequency of vestibular syndromes and vertigo, specifically in the different stroke subtypes (atherothrombotic infarcts, cardioembolic infarcts, lacunar stroke, infarcts of unusual etiology, and infarcts of unknown etiology or intracerebral hemorrhage), are lacking. However, this clinical aspect is potentially relevant due to the fact that the pathophysiology, prognosis, and clinical features of ischemic small vessel strokes are different from all other cerebral infarcts In addition, small demyelinating plaques or small lacunar infarcts in the root entry zone and/or in the vestibular nuclei may mimic a vestibular neuritis [28,29,30,31], which combines rotational vertigo and spontaneous nystagmus with abolition of caloric responses on the affected side. These are combined, in turn, with a masseteric paresis manifested by the masseter reflex or by ocular motor abnormalities such as saccadic pursuit [32].

Likewise, it is important to know the early clinical outcome in acute stroke patients. In previous clinical studies, older age, atrial fibrillation, NIHSS scale, presence of previous stroke, altered consciousness, and cerebral hemorrhage, but not the presence of vertigo, were predictors associated with both in-hospital mortality and hospital stay >12 days [2,33]. However, in one study, vertigo was a clinical symptom of hemorrhagic stroke of infratentorial location with a poor outcome [34].

The main objective of this single-center comparative clinical study was to expand and update the knowledge on the poorly understood relationship between vertigo and acute stroke, mainly in relation to the neurological clinical profile, the specific cerebral and vascular location, its frequency in the different stroke subtypes, and the association between vertigo, early evolution, and prognosis.

## 2. Materials and Methods

Since 1986, the Hospital Universitari Sagrat Cor (a 350-bed teaching hospital in Barcelona, Catalonia, Spain, serving a population of more than 300,000 inhabitants) established a hospital-based stroke registry [35,36]. Data for all patients included in our stroke registry were entered following a standardized protocol with 186 items (demographic data, risk factors, clinical characteristics, laboratory and neuroimaging data, complications, and outcomes). The use of the same protocol for all patients ensured the integrity of the information in the database. Stroke subtypes were classified according to the criteria of the Cerebrovascular Study Group of the Spanish Society of Neurology [37], which are similar to the classification of the National Institute of Neurological Disorders and Stroke [38]. The study protocol was approved by the Clinical Research Ethics Committee of the Hospital.

A total of 4600 consecutive acute stroke patients were screened for the study. The frequency of the different stroke subtypes in the acute stroke registry was as follows: 957 cardioembolic infarcts (20.8%), 946 atherothrombotic infarcts (20.5%), 865 lacunar strokes (18.8%), 128 cerebral infarctions of unusual cause (2.8%), 374 cerebral infarctions of unknown cause (8.1%), 761 transient ischemic attacks (16.5%), 473 intracerebral hemorrhages (10.3%), 52 subarachnoid hemorrhages (1.1%), and 44 spontaneous subdural hemorrhage/spontaneous epidural hemorrhages (1%).

For the purposes of the study, only patients diagnosed with cardioembolic infarction, atherothrombotic infarct, lacunar stroke, cerebral infarction of unusual cause, cerebral infarction of unknown cause, and intracerebral hemorrhage were selected from the stroke registry database (Figure 1).

The criterion for classifying patients with vertigo was the definition from the Bárány Society: “the sensation of self-motion when no self-motion is occurring or the sensation of distorted self-motion during an otherwise normal head movement” [39]. All patients were admitted within 48 hours of symptom onset. On admission, demographic data and the most salient clinical and neurological symptoms were recorded, as well as the results of the different laboratory tests (blood cell count, biochemical profile, serum electrolytes, and urinalysis), chest X-ray, 12-lead electrocardiogram, and brain CT (96.5%) and/or MRI (38.7%), together with the clinical investigations performed at the discretion of the responsible neurologist. The registry included medical complications—respiratory, cardiac, urinary, renal, and vascular—and mortality during the acute phase of the disease. The degree of clinical disability at hospital discharge was assessed according to the modified Rankin scale (mRS) [40].

Demographic characteristics of the patients included in the study are reported in Table 1. 

A comparative analysis was performed between patients with presence or absence of vertigo. Continuous data were summarized as mean and standard deviation (SD) and categorical data as frequencies and percentages. The distributions of variables in patients in both groups were compared with the chi-square (χ^2^) test or the Fisher’s exact test for categorical variables, and the Student’s t test for quantitative variables. For all analyses, *p* < 0.05 was considered to indicate significance. Covariates with a *p* value < 0.20 in the univariate test were entered into a multivariate logistic regression model with a stepwise selection method, in which the presence of vertigo (versus absence of vertigo) was the dependent variable. The model was based on demographic data, cardiovascular risk factors, clinical characteristics, and cerebral and vascular topography. 

To establish statistically significant criteria associated with both the presence and absence of vertigo, the odds ratio (OR), and the 95% confidence interval (CI) were calculated. The receiver operating characteristic (ROC) curve was used to assess the accuracy of the model for identifying vertigo in acute stroke; sensitivity, specificity, and positiveand negative predictive values were calculated. Statistics were analyzed using the package SPSS (Version 20 for Mac; SPSS Inc., Chicago, IL, USA).

## 3. Results

### 3.1. General Data

The study population included 3743 consecutive patients diagnosed with acute ischemic stroke or spontaneous intracerebral hemorrhage. Vertigo and laberynthine symptoms at stroke onset were present in 149 patients (3.9%) with a mean age of 71.8 ± 12.9 years. The remaining 3594 patients without vertigo were significantly older, with a mean age of 75.9 ± 11.6 years. The percentage of women was similar in the vertigo group (47.6%) compared to the non-vertigo group (51.1%) (Table 1).

### 3.2. Differences between the Vertigo and the Non-Vertigo Acute Stroke Groups

The presence of vertigo differed in the different stroke subtypes and was more frequent in atherothrombotic infarction (34.7%), intracerebral hemorrhage (19.7%), cardioembolic stroke (18.4%), and infarction of unknown cause (15%), and was less frequent in lacunar infarcts (9.5%) and infarction of unusual cause (2.7%).

The results of the differences between the vertigo and the non-vertigo groups by univariate analysis are presented in Table 2. Overall, hyperlipidemia, headache, nausea or vomiting, ataxia, and cranial nerve palsy were significantly more frequent in the vertigo group, whereas among those aged 85 years or older, atrial fibrillation (AF), nephropathy, limb weakness, and speech disturbances were significantly more frequent in the non-vertigo group. 

The distribution of lesions according to the cerebral location in the medulla, pons, and cerebellum, as well as vascular topography in the vertebral artery, basilar artery, posteroinferior cerebellar artery, anteroinferior cerebellar artery, and superior cerebellar artery were more frequent in the vertigo group.

Early outcome was similar in the vertigo group compared with the non-vertigo group, with a similar percentage of patients free of symptoms and with a mild neurological deficit at hospital discharge. In addition, the in-hospital mortality rate was not significantly higher in the vertigo group. 

### 3.3. Multivariate Analysis

The results of the multivariate analysis are shown in Table 3. The logistic regression model based on demographics and cardiovascular risk factors, clinical characteristics, brain location, and vascular topography reported that cerebellar location (OR 5.59), presence of nausea or vomiting (OR 4.48), medulla location (OR 2.87), pons location (OR 2.39), basilar artery (OR 2.36), ataxia (OR 2.33), and headache (OR 2.31) were independently associated with the vertigo group, whereas speech disturbances (OR 0.63) and limb weakness (OR 0.47) were independent variables associated with patients without vertigo. According to this model, cases of acute stroke with vertigo versus non-vertigo were correctly classified in 84.4% of the cases.

Figure 2 shows the ROC curve of the accuracy of the regression model. The area under the curve (AUC) was 0.857. The sensitivity was 73%, the specificity was 85%, the positive predictive value was 16%, and the negative predictive value was 99%.

## 4. Discussion

The main causes of vertigo are peripheral, such as Ménière’s disease and vestibular migraine [27]. However, vertigo can be associated with acute stroke. In this study, 149 patients presented vertigo associated with acute cerebrovascular disease, with a prevalence of 3.9%: a percentage similar to that of other clinical studies [41,42,43].

The study of vertigo in acute stroke has usually been performed in patients selected by vascular topography, such as vertebrobasilar stroke, ischemic syndromes, or cerebral location such as brainstem and cerebellar strokes [44,45], whereas studies of acute strokes analyzed globally from stroke databanks, such as our study, are scarce. A study conducted at Hasan Sadikin General Hospital in Indonesia [43] also sought to determine the profile of stroke patients with vertigo in a retrospective clinical analysis and found that stroke patients experiencing vertigo were more likely to be women (59%), in contrast to our results in which no significant gender-related differences were obtained.

To our knowledge, our study is the only study conducted in a stroke database with the main objective of analyzing the clinical relevance of vertigo in acute stroke, and our sample is one of the largest to date with the aim of studying the clinical, topographic, and prognostic predictors of vertigo in acute stroke. In the present work, we have shown that the presence of vertigo is more frequent in non-lacunar acute stroke subtypes (atherothrombotic infarct, cardioembolic infarct, and intracerebral hemorrhage) compared to patients with lacunar infarcts, which are the subtype of ischemic stroke with a characteristic small lesional cerebral size (usually less than 15 mm maximum diameter) [46,47,48,49,50]. These results are similar to those of other studies in which an inverse relationship between volume and vertigo was demonstrated, with significantly less frequent presence of vertigo in small-sized acute strokes [51,52,53,54,55,56]. We speculate that larger cerebral infarcts or hemorrhages mediate the development of vascular vertigo through involvement of various posterior infratentorial brain structures and interconnections related to central vestibular connections.

As in previous studies, we found that vertebrobasilar stroke was significantly associated with the development of vertigo [44,57]. Tao et al. [58] reported a higher frequency of vertigo in vertebrobasilar strokes compared to anterior circulation strokes (18.9% vs. 1.7%). In relation to vascular topography, it should be noted that the basilar artery (24.5%), vertebral artery (14.3%), superior cerebellar artery (12.2%), posteroinferior cerebellar artery (10.9%), and anteroinferior cerebellar artery (4.1%) were the specific vascular arteries which were significantly more frequent in the group presenting with acute stroke and vertigo. This is in contrast to the results of other authors who differentiated only the vertebrobasilar arterial territory versus the carotid topography without analyzing—as we did in our study—the different cerebral arteries arising from the vertebrobasilar vascular territory in detail [22,25,41,59]. 

The study of Doijiri et al. [60] stated that the posteroinferior cerebellar artery was the most frequently associated with vertigo and stroke. Lee et al. [45] also found that the posteroinferior cerebellar artery was the most frequently disrupted, followed by the superior cerebellar artery with vertiginous symptomatology.

In our study, the presence of vertigo predicts the cerebral location of the acute ischemic or hemorrhagic stroke mainly in the cerebellum and brainstem. The development of vascular vertigo is associated with cerebrovascular lesions affecting the following central vestibular structures: the vestibular nuclei in the dorsolateral portion of the rostral medulla, the nucleus prepositus hypoglossi in the dorsal brainstem, and the dorsal insular cortex, as well as the cerebellar tonsil, the flocculus, the nodulus, and the inferior cerebellar peduncles [61]. The brainstem contains the neural structures involved in the integration and transmission of vestibular signals; therefore, lesions of the brainstem and cerebellum result in various vestibular symptoms and signs [62]. 

However, the clinical spectrum of cerebellar strokes rarely presents with isolated dizziness or without clear central neurological deficits (e.g., dysarthria or ataxia). Sometimes, isolated acute labyrinthine damage may herald impending pontocerebellar involvement in infarction in the territory of the anterior inferior cerebellar artery and may cause brainstem or cerebellar strokes; vestibular strokes are often overlooked because they mimic more benign hearing disorders [24,63]. Lee et al. [64] and Deng et al. [65] found that vestibular structures were more vulnerable to ischemia than any other structures in the brainstem and cerebellum, with the medial vestibular nucleus being the most vulnerable.

The unusual association of lacunar stroke with vertigo could be due to the usual lesional topography of lacunar strokes in the centrum semiovale, internal capsule, basis pontis, and ventroposterolateral thalamic nucleus, which are all cerebral structures that are remote from the central vestibular connecting pathways [66,67,68]. 

Furthermore, the odds ratio for the development of vertigo shows a clinical profile with three main associated clinical features: presence of nausea or vomiting, ataxia, and headache. These are all caused by the involvement of vascularized brain structures of the vertebrobasilar system. It should be noted that differentiation between brainstem and cerebellar lesions is, in most clinical cases of acute stroke, impossible because the major infratentorial arteries supply both the brainstem and the cerebellum [27]. 

Our results are consistent with those of Harriott et al. [69] and Levedova et al. [70], who found that headache at the onset of acute stroke is more frequent in posterior circulation stroke than in the carotid arterial territory. Additionally, it is noteworthy that the presence of speech disturbances and limb weakness is not associated with the vertigo group. These results agree with those of Elhfnawy et al. [51] who, in their study, found that there were fewer patients with associated neurological deficits in the vertigo group. We must differentiate our work from the interesting study published by Qiu et al. [71]. In their study, previous vertigo attacks were a risk factor or predictor of the presence of posterior acute ischemic stroke, whereas, in our study, vertigo and labyrinthine symptoms are clinical symptoms present at the onset of stroke in all patients analyzed. 

Insular acute stroke is a rare and underreported pathology, and its clinical presentation is heterogeneous, although patients with acute insular stroke may also present a vestibular-like syndrome with isolated "vertigo" or "dizziness" with instability [72]. In our sample, we did not find any patient with this eventuality. However, it should be noted that, among anterior circulation strokes, isolated insular strokes are the most representative due to dysfunction of a relevant hub of the vestibular cortical network [72].

Little is known about the early outcome of patients with vestibular loss and vertigo associated with acute stroke. Importantly, acute cerebellar stroke may develop mass effect and intracranial hypertension. Although large cerebellar strokes can cause brainstem compression, resulting in hydrocephalus, cardiorespiratory complications, cerebral herniation, coma, and death [2,36,73,74], in our study, patients in the vertigo group had a similar percentage of patients free of symptoms and with mild neurological deficits at hospital discharge as the non-vertigo group. In addition, the in-hospital mortality rate was not significantly higher in either group. This negative association between vertigo and outcome is probably due in part to the fact that neurovegetative symptoms are very eloquent clinically, and, thus, we speculate that these patients are evaluated earlier in the emergency department and can receive the therapeutic regimens of choice in the most acute phase adequately. 

From our results, we deduce that the presence of vertigo in patients with acute stroke is relevant to cerebral localization in cerebellum and brainstem (medulla and pons), as well as vertebrobasilar vascular topography, mainly in the territory of basilar artery; however, this does not translate into prognostic or early outcome significance. This study has several limitations. First, it is a retrospective, cross-sectional clinical study conducted at a single center. This design may have contributed to selection bias. In this regard, we recommend the performance of future large multicenter studies. Likewise, further validation of our results in a new prospective clinical study would be of interest. Secondly, the study did not consider the different vascular segments of the cerebral arteries (e.g., M1, M2, or M3 segments in MCA ischemic stroke), nor did it consider other specific partial brain locations such as insular topography. Another limitation of the study would be to not have analyzed the prognostic value of the presence of vertigo in acute stroke in the medium or long term. 

In future studies, these aspects would be interesting lines of research. However, the assessment of the methods used in this study based on the results of a stroke data bank from a large sample of 4600 consecutive patients collected over a 24-year period is more or less objective. Additionally, a future line of investigation would be the use of high-resolution magnetic resonance imaging, which would allow for a better understanding between different vascular segments or small topographic brain locations and vertigo in acute stroke patients.

## 5. Conclusions

In conclusion, acute central vestibular syndrome is a strong predictor of cerebellar or dorsal brainstem acute stroke (pons or medulla involvement) and central vascular vertigo is more related to the larger volume cerebrovascular non-lacunar subtypes: atherothrombotic and cardioembolic infarcts, and spontaneous intracerebral hemorrhage. The presence of vertigo in acute stroke is associated with vertebrobasilar arterial location, mainly in basilar artery involvement.

Consequently, if a stroke occurs in the cerebellum or brainstem, areas that control balance in the brain, the patient may suffer vertigo. However, vertigo in the acute phase of stroke did not result in higher in-hospital mortality or worse early clinical outcomes. Possibly, since vertigo is a very eloquent clinical symptom, early treatment is more commonly carried out in acute stroke patients with vertigo, causing this early therapeutic assessment to be a probable clinical benefit.

## Figures and Tables

**Figure 1 biomedicines-10-02830-f001:**
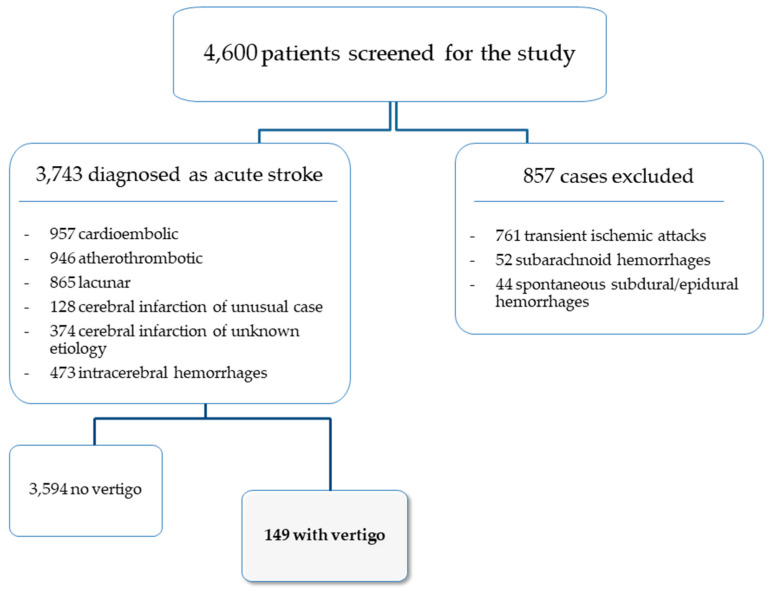
Flow-chart of patients included in the study.

**Figure 2 biomedicines-10-02830-f002:**
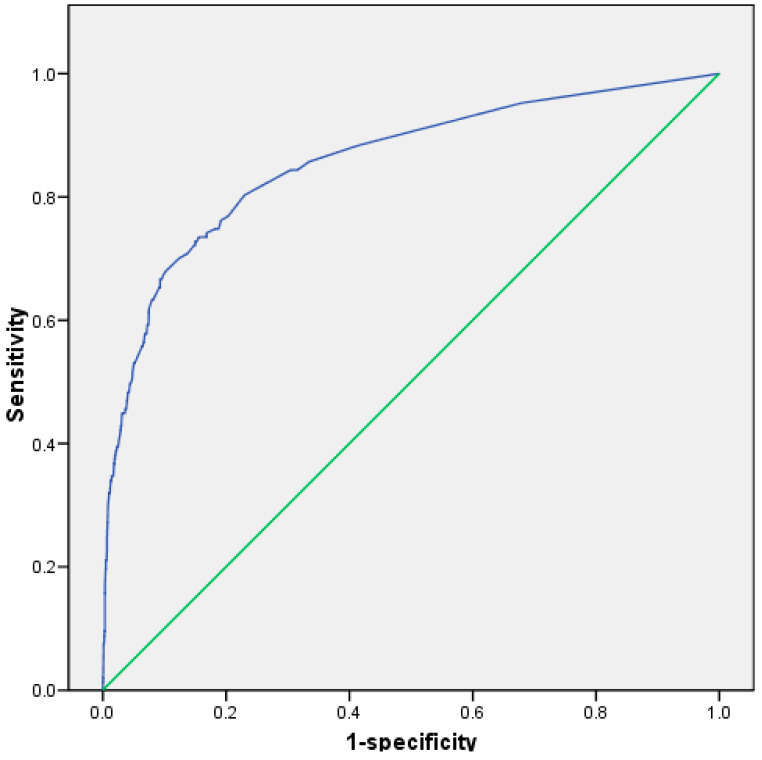
ROC curve for regression model including demographics, cardiovascular risk factors, clinical characteristics, and cerebral and vascular topography (AUC: 0.857). AUC: area under the curve; ROC: receiver operating characteristics.

**Table 1 biomedicines-10-02830-t001:** Demographics, data in acute stroke patients with vertigo versus non-vertigo.

Variable	Acute Stroke with Vertigo *n* = 149	Acute Stroke without Vertigo *n* = 3594	*p* Value
Age, years, mean (SD)	71.8 (12.9)	75.9 (11.6)	0.0001
Age ≥ 85 years old	21 (14.3)	827 (23)	0.013
Sex			0.676
Males	79 (52.6)	1757 (49.0)	
Females	70 (47.6)	1837 (51.1)	

**Table 2 biomedicines-10-02830-t002:** Cerebrovascular risk factors, neuroimaging, and outcomes in acute stroke patients with vertigo versus non-vertigo.

Variable	Acute Stroke with Vertigo *n* = 149	Acute Stroke without Vertigo *n* = 3594	*p* Value
Risk factors			
Hypertension	96 (65.3)	2094 (58.3)	0.089
Atrial fibrillation	30 (20.4)	1024 (28.5)	0.033
Hyperlipidemia	38 (25.9)	664 (18.5)	0.025
Diabetes mellitus	34 (23.1)	826 (23.0)	0.517
Ischemic heart disease	22 (15.0)	537 (14.0)	0.534
Heavy smoking (>20 cigarettes/day)	22 (15.0)	372 (10.4)	0.074
Chronic obstructive pulmonary disease	7 (4.8)	320 (8.9)	0.081
Nephropathy	0 (0.0)	137 (3.8)	0.016
Clinical findings			
Headache	58 (39.5)	442 (12.3)	0.0001
Nausea, vomiting	68 (46.3)	260 (7.2)	0.0001
Limb weakness	71 (48.3)	2728 (75.9)	0.0001
Speech disturbances (dysarthria, aphasia)	48 (32.7)	1821 (50.7)	0.0001
Ataxia	54 (36.7)	204 (5.7)	0.0001
Cranial nerve palsy	25 (17.0)	174 (4.8)	0.0001
Neuroimaging findings topography			
Frontal lobe	9 (6.1)	505 (14.1)	0.006
Parietal lobe	14 (9.5)	873 (24.3)	0.0001
Temporal lobe	15 (10.2)	884 (24.6)	0.0001
Internal capsule	11 (7.5)	648 (18.0)	0.001
Basal ganglia	10 (6.8)	503 (14)	0.013
Midbrain	6 (4.1)	45 (1.3)	0.011
Pons	27 (18.4)	185 (5.1)	0.0001
Medulla	14 (9.5)	35 (1.0)	0.000
Cerebellum	46 (31.3)	83 (2.3)	0.0001
Middle cerebral artery	26 (17.7)	1921 (53.5)	0.0001
Vertebral artery	21 (14.3)	101 (2.8)	0.0001
Basilar artery	36 (24.5)	226 (6.3)	0.0001
Posteroinferior cerebellar artery	16 (10.9)	20 (0.6)	0.0001
Anteroinferior cerebellar artery	6 (4.1)	11 (0.3)	0.0001
Superior cerebellar artery	18 (12.2)	32 (0.9)	0.0001
Stroke subtypes			0.0001
Atherothrombotic infarct	51 (34.7)	894 (24.9)	
Cardioembolic infarct	27 (18.4)	930 (25.9)	
Infarction of unknown cause	22 (15)	352 (9.8)	
Lacunar stroke	14 (9.5)	850 (23.7)	
Infarctions of unusual cause	4 (2.7)	124 (3.5)	
Intracerebral hemorrhage	29 (19.7)	444 (12.4)	
Outcome			
Symptom-free at discharge	21 (14.3)	352 (15.4)	0.723
In-hospital death	17 (11.6)	523 (14.6)	0.312
Length of stay, days, median (interquartile range)	12 (8–20)	12 (8–20)	0.977
Prolonged hospital stay > 12 days	65 (44.2)	1695 (47.2)	0.269

Data expressed as numbers and percentages in parenthesis.

**Table 3 biomedicines-10-02830-t003:** Results of multivariate analysis: variables independently associated with vertigo in acute stroke patients.

Regression Model	Coefficient (*β*)	Standard Error	Odds Ratio (95% Confidence Interval)	*p* Value
Model based on demographics, risk factors, clinical characteristics, and cerebral and vascular topography				
Cerebellum	1.721	0.278	5.59 (3.24–9.64)	0.0001
Nausea, vomiting	1.500	0.214	4.48 (2.95–6.82)	0.0001
Medulla involvement	1.055	0.401	2.87 (1.35–6.30)	0.009
Pons involvement	0.870	0.325	2.39 (1.26–4.51)	0.007
Basilar artery involvement	0.858	0.291	2.36 (1.33–4.17)	0.003
Ataxia	0.846	0.256	2.33 (1.41–3.85)	0.001
Headache	0.836	0.211	2.31 (1.53–3.49)	0.0001
Speech disturbances	−0.457	0.203	0.63 (0.42–0.94)	0.025
Limb weakness	−0.755	0.196	0.47 (0.32–0.69)	0.0001

Hosmer-Lemeshow goodness-of-fit test χ^2^ = 1.070, df = 4; *p* = 0.0899; vertigo versus non-vertigo acute stroke subjects were correctly classified in 84.4% of cases.

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
