# Peer review of "Vertigo in Acute Stroke Is a Predictor of Brain Location but Is Not Related to Early Outcome: The Experience of Sagrat Cor Hospital of Barcelona Stroke Registry"

_biomedicines, 2022, doi:10.3390/biomedicines10112830_

Round 1

Reviewer 1 Report

The Authors describe their experience with acute stroke and the relationship between vertigo and stroke. The study is well readably, well-conducted, and according to the literature. In this comparative analysis between stroke patients with or without vertigo, it could be helpful to know if the labyrinthine symptoms were preexistent and the timing with stroke. If the Authors have this information would be useful if they add it to the text.

Author Response

We appreciate your supportive comment regarding the scientific interest of the study.

We added and clarified the information related to the timing of vertigo in the study sample in the text. “Vertigo and the laberynthine symptoms at stroke onset” (Results, page 5 lines 201-202).

We thank the reviewer for his time and helpful suggestions to improve the quality of the manuscript.

Reviewer 2 Report

This interesting study supports the idea that cerebral topography of vascular lesions may influence the presence of vertigo. It is well-conducted and very interesting, providing useful insight on vestibular symptoms in ischemic stroke. However, I have some considerations:

1.     Lines 31-41. I suggest to move this paragraph after line 55.

2.     Lines 76-78. It is correct, but lesion studies exploring vertigo must be cited (see comments 3-4-5).

3.     Lines 80-81. “vertigo has also long been recognized as an isolated manifestation of anterior inferior cerebellar artery ischemic stroke”. It is not complete alone. Moreover, small insular strokes have been reported to cause isolated vertigo in many cases. I suggest to discuss this peculiarity, as it is, in my opinion, the only location able to cause isolated vestibular syndromes. Are there any cases in this sample?

4.     A sufficient emphasis should be given to the role of posterior insular cortex in the onset of vertigo, in accord with lesion studies. Indeed, strokes in the right insula very often cause isolated vertigo (33%) due to lateralization of vestibular functions; of interest, lesion studies seem to point out that the right PIC might serve as a relevant hub for vestibular functions, as strokes in the right PIC more frequently cause a “vestibular-like syndrome” (50%) compared to the left PIC (20%) (Di Stefano et al, 2021. Clinical presentation of strokes confined to the insula: a systematic review of literature). Are there any cases of insular stroke among these 143 patients? Any evidence of lateralization? 

5.     I suggest to discuss the fact that, among anterior circulation strokes, isolated insular strokes are the more representative due to the dysfunction of a relevant hub in the vestibular cortical network.

6.     Table 1. MCA strokes probably include insular strokes (M2). MCA strokes should be classified in M1-M2-M3 and isolated small strokes should be considered differently due to the very large ischemic territory. This should be added as a limitation.

7.     Lines 232-233. “our sample is the largest to date with the aim of studying the 232 clinical, topographic and prognostic predictors of vertigo in acute stroke.” I am not sure it is the largest sample, hence I suggest to reduce emphasis on this study.

8.     There are no relevant grammar issues. 

Author Response

Thank you for your valuable comments and suggestions.

  1. We have corrected this part and moved the paragraph  62-72 (former paragraph 31-41) after  former line 55, line 62 in the revised manuscript.
  2. We have cited the studies reporting vertigo: [24-27] (page 3, line 96)
  3. We added on page 3 (lines 99-100) that small insular strokes have been reported to cause isolated vertigo. However, in our sample we have not found any patients with this unusual eventuality
  4. In accordance with the reviewer’s suggestion, in the revised version of the manuscript, we have added this sentence in the Discussion: “Insular acute stroke is a rare and underreported pathology and its clinical presentation is heterogeneous, although patients with acute insular stroke may also present a vestibular-like syndrome with isolated "vertigo" or "dizziness" with instability [72]. In our sample we did not find any patient with this eventuality.”(page 10, lines 333-336).

The reference suggested by the reviewer is also included:

Reference 72: Di Stefano, V., De Angelis, M.V., Montemitro, C. et al.Clinical presentation of strokes confined to the insula: a systematic review of literature. Neurol Sci, 2021, 42, 1697–1704. https://doi.org/10.1007/s10072-021-05109-1

5.In the Discussion, we have added the sentence: “among anterior circulation strokes, isolated insular strokes are the more representative due to the dysfunction of a relevant hub of the vestibular cortical network” (page 10, 337-338, and reference 72)

6. We have expanded the text by adding a limitation of our study by not considering different vascular segments of the cerebral arteries (example: M1, M2 or M3 segments in MCA ischemic stroke) or other specific cerebral locations as an isolated insular topography  (page 11, lines 361-362).

7.We have reduced the emphasis in this study.  We have changed: “our sample is the largest” by “our sample is one of the largest to date” (page 9, line 267)

We thank the reviewer for his time and helpful suggestions to improve the quality of the manuscript.

Reviewer 3 Report

The current manuscript is impressive that it co-relates between vertigo and stroke. But similar type of work has been published before. 

The author needs to differentiate their work form the published articles. 

for eg; The research article published in 2020 has similar interest as the current manuscript. 

ORIGINAL RESEARCH article

Front. Neurol., 16 December 2020
Sec. Stroke
https://doi.org/10.3389/fneur.2020.593524

New Insights Into Vertigo Attack Frequency as a Predictor of Ischemic Stroke.

The author can put some efforts to validate the study based on the multiple stokes patients suffering with vertigo and without vertigo and with onetime stroke.

Author Response

We wish to thank the reviewer for the interest you expended on our behalf and his supporting comments.

  1.  We have added and clarified this information in the text (Discussion page 10, line 329-332): “We must differentiate our work from the interesting study published by Qiu et al [71]. In their study, previous vertigo attacks were a risk factor or predictor of the presence of posterior acute ischemic stroke, whereas in our study, vertigo and labyrinthine symptoms are clinical symptoms present at the onset of stroke in all patients analyzed”.

The reference suggested by the reviewer is also included: 

Reference 71 Qiu D, Zhang L, Deng J, Xia Z, Duan J, Wang J and Zhang R. New Insights Into Vertigo Attack Frequency as a Predictor of Ischemic Stroke. Front. Neurol.2020, 11, 593524. doi: 10.3389/fneur.2020.593524

2.We have added in the text as a future line of research “further validation of our results in a new prospective clinical study would be of interest” (Discussion page 11, line 358-360)